# Global burden and future projections of non-communicable diseases (2000–2050): Progress toward SDG 3.4 and disparities across regions and risk factors

Omar Freihat[1]*, Dvid Sipos[2], Maria Aamir[1], Arpad Kovacs[3]

**1** Department of public health, Abu Dhabi University, Abu Dhabi, UAE, **2** College of Health science, University of Pecs, Pecs, Hungary, **3** Department of Oncoradiology, University of Debrecen, Debrecen, Hungary

* Omar.freihat@adu.ac.ae

## Abstract

### Background

Non-communicable diseases (NCDs), such as cardiovascular diseases (CVDs), neoplasms, chronic respiratory diseases (CRDs), and diabetes mellitus (DM), are the leading cause of death globally and a major health challenge, threatening Sustainable Development Goal (SDG) 3.4 to reduce premature mortality by one-third by 2030. The study aimed to quantify the current and projected burden of these NCDs, assess disparities, and identify prevention opportunities.

### Methods

Using data from the Global Burden of Disease Study (GBD) 2021 by the Institute for Health Metrics and Evaluation, the study analysed incidence, mortality, and disability-adjusted life years (DALYs) for CVDs, neoplasms, CRDs, and DM from 2000 to 2021, with projections to 2050. Trends were stratified by time, WHO region, Socio-Demographic Index (SDI), gender, and country, alongside risk factor contributions (metabolic, behavioural, environmental). Age-standardized rates and average annual percent changes (AAPC) were calculated to assess trajectories.

### Results

In 2021, NCDs accounted for 12·4 billion incident cases (rate: 156,680·64 per 100,000), 43·8 million deaths (64·5% of total, rate: 554·63 per 100,000), and 1·73 billion DALYs (59·9%, rate: 21,887·11 per 100,000). CVDs led with 19·4 million deaths (28·6%), followed by neoplasms (9·9 million, 14·6%), CRDs (4·4 million, 6·5%), and DM (1·7 million, 2·4%). From 2000–2021, DM incidence rose sharply (AAPC: 2·41%), while CVD mortality increased slightly (AAPC: 0·21%). High SDI regions showed

**Data availability statement:** All relevant data are within the manuscript and its Supporting Information files. Data can be found https://ihm.org/.

**Funding:** The author(s) received no specific funding for this work.

**Competing interests:** he authors have declared that no competing interests exist.

peak DM incidence (461·41 per 100,000), Europe led CVD mortality (419·62), and males had higher CVD deaths (258·55 vs 233·41). Projections estimate 75·5 million deaths and 2·44 billion DALYs by 2050, driven by CVDs (86·1% of deaths). Metabolic risks like hypertension dominated CVD burdens.

## Conclusions

The NCD burden's rapid rise, marked disparities, and projected escalation demand urgent, equity-focused prevention. Tailored strategies, targeting metabolic risks, gender gaps, and regional inequities, are critical to achieve SDG 3.4 and mitigate this global crisis by 2050.

---

## Introduction

Non-communicable diseases (NCDs), such as cardiovascular diseases, cancers, chronic respiratory conditions, and diabetes, are the predominant cause of death globally, accounting for approximately 74% of all mortality, as reported by the World Health Organization [1]. This escalating burden, intensifying over recent decades, places significant pressure on health systems worldwide, particularly in low- and middle-income countries (LMICs), where limited resources and infrastructure exacerbate challenges [2]. NCDs disproportionately affect the most vulnerable socioeconomic communities, including those in LMICs, due to factors such as poverty, limited healthcare access, and social inequities, rather than solely rapid urbanization or aging populations [3]. In LMICs, this reflects an epidemiological transition from infectious diseases to chronic conditions, compounding existing health challenges [4,5].

Several interconnected factors fuel the rise of NCDs, including sedentary lifestyles, poor nutrition, tobacco use, excessive alcohol consumption, and environmental exposures such as air pollution [6]. These risk factors are amplified by socioeconomic inequalities, which limit access to prevention, early detection, and treatment, particularly in resource-constrained settings [7]. While high-income countries have made strides in mitigating NCD prevalence through robust public health policies, LMICs continue to grapple with inadequate funding and systemic barriers, resulting in higher morbidity and economic costs [8].

This paper examines the global burden of NCDs by analysing trends in incidence, mortality, and disability-adjusted life years (DALYs) across regions and income levels from 2000 to 2021, with projections extending to 2050. Drawing on data from the Global Burden of Disease (GBD) study and WHO reports, the study employs statistical modelling to assess historical patterns and forecast future trajectories under different policy scenarios. It also investigates the influence of key risk factors such as obesity, smoking, and pollution on NCD outcomes, aiming to identify priorities for intervention. By providing a detailed synthesis of past trends and future projections, this research offers actionable insights for policymakers and health professionals seeking to address the mounting NCD crisis.

## Materials and methods

We conducted a secondary analysis of the global burden of non-communicable diseases (NCDs), specifically cardio-vascular diseases (CVD), neoplasms, diabetes mellitus (DM), and chronic respiratory diseases (CRDs), from 2000 to 2021, with projections to 2050. These NCDs were classified using the Global Burden of Disease (GBD) 2019 frame-work, which organizes diseases into three levels: Level 1 includes all NCDs as a broad group, and Level 2 includes major categories such as CVDs, neoplasms, DM, and CRDs [9], S0 Table in S1 File. Our study focused on these Level 2 categories without analysing specific Level 3 subcategories. Data for incidence, mortality, and disability-adjusted life years (DALYs) were sourced from the GBD 2021 estimates provided by the Institute for Health Metrics and Evaluation (IHME), covering 204 countries and territories. The use of the GBD 2019 framework ensures standardized disease classification and methodological consistency, while the updated GBD 2021 dataset provides the most recent esti-mates, including risk factor contributions and projections to 2050 based on historical trends. [9], covering 204 countries and territories. This comprehensive country selection aligns with the GBD 2019's standardized framework, which uses consistent data collection and reporting methods across nations to ensure comparability with global health metrics stan-dards. The GBD 2019 provides estimates of incidence, mortality, and disability-adjusted life years (DALYs) [9], a metric combining years of life lost and years lived with disability, derived from integrated health data, along with risk factor contributions and 2050 projections based on historical trends. Ethics approval and consent to participate is not applica-ble due to the nature of the study.

We reported age-standardized incidence rates (new cases per 100,000 population per year), mortality rates (deaths per 100,000 population per year), and disability-adjusted life years (DALYs; per 100,000 population per year), which combine years of life lost and years lived with disability, for cardiovascular diseases (CVD), neoplasms, diabetes mellitus (DM), and chronic respiratory diseases (CRDs) [9]. Outcomes were evaluated globally, by Socio-Demographic Index (SDI; Low, Low-Middle, Middle, High-Middle, High) [9], WHO region (African, Americas, Eastern Mediterranean, European, South-East Asia, Western Pacific), sex, and country, with highest and lowest rates highlighted where applicable.

The 2021 global NCD burden was aggregated from GBD (rates per 100,000), with proportions calculated as percent-ages of total deaths or DALYs. Trends from 2000 to 2021 were quantified using the average annual percentage change (AAPC), calculated as [10]:

$$\text{AAPC} = \left( \left( \frac{\text{Rates in 2021}}{\text{Rates in 2000}} \right)^{\frac{1}{21}} - 1 \right) * 100\%$$

Where values represent ASR (incidence), deaths (mortality), or DALYs. We note that using counts may partly capture demographic effects such as population growth and aging, rather than solely reflecting changes in disease risk. To address this, country, SDI, and WHO regional comparisons were complemented with age-standardized rates provided by GBD, particularly for the 2050 projections. Risk factor impacts were reported as GBD-provided deaths and DALYs per 100,000 population.

## Results

In alignment with the WHO's Global NCD Monitoring Framework and the Sustainable Development Goal (SDG) 3.4 and WHO report 2024 recommendations of NCDs, aiming to reduce premature mortality from non-communicable diseases by one-third by 2030, this study conducted focused analyses on the four major NCDs: cardiovascular diseases, neoplasms, chronic respiratory diseases, and diabetes mellitus. These diseases were examined in terms of incidence, mortality, and disability-adjusted life years (DALYs), with analyses stratified by time, region, and socio-demographic context. In addi-tion to evaluating historical trends, projections were generated through 2050 to estimate the future burden of NCDs and assess alignment with global health targets.

## Global burden of NCD's in 2021

In 2021, the total global incidence of non-communicable diseases (NCDs) was estimated at 12.36 billion cases, with an overall rate of 156,681 per 100,000 population. The largest contributors to this burden were other non-communicable diseases (5.20 billion cases) and skin and subcutaneous diseases (4.69 billion cases), followed by neurological, digestive, and mental disorders. Our study specifically focused on four major NCD categories central to SDG target 3.4: cardiovascular diseases (CVDs), neoplasms, chronic respiratory diseases (CRDs), and diabetes mellitus (DM). Collectively, these conditions accounted for approximately 233 million new cases, representing about 1.9% of the total NCD incidence worldwide. Within this group, CVDs contributed 66.8 million cases (0.54% of total incidence), neoplasms 66.5 million (0.54%), CRDs 55.2 million (0.45%), and DM 44.9 million (0.36%). Regarding mortality, NCDs were responsible for 43,768,182 deaths, representing 64·5% of all deaths worldwide, with a death rate of 554·63 per 100,000. CVDs were the leading cause of death, resulting in 19,414,853 fatalities (28·6% of total deaths), followed by neoplasms with 9,888,413 deaths (14·6%), CRDs with 4,414,182 deaths (6·5%), and DM with 1,656,635 deaths (2·4%). Collectively, these four conditions, central to SDG target 3.4, accounted for approximately 37.6 million deaths, representing 86% of all NCD mortality worldwide.

Among all categories, the largest contributors were cardiovascular diseases (428.3 million DALYs) and neoplasms (253.3 million), followed by chronic respiratory diseases (108.5 million) and diabetes mellitus (123.7 million). Collectively, these four major conditions accounted for 913.8 million DALYs, representing more than half (52.9%) of the global NCD DALY burden. Within this group, CVDs alone accounted for nearly one-quarter of NCD DALYs (24.8%), with neoplasms (14.7%), diabetes (7.2%), and CRDs (6.3%) contributing substantial shares. **S1, S1A in** S1 File.

## SDI and WHO region analysis of NCDs (2021)

The burden of NCDs varied widely across SDI groups **(S1, S1B in** S1 File, Fig 1A). Incidence rates for diabetes, CVDs, and neoplasms were consistently highest in high-SDI regions and lowest in low-SDI regions. CRDs showed a different pattern, with elevated incidence in both high- and low-SDI regions, but lower levels in middle-SDI groups. Mortality also varied: diabetes deaths were greatest in middle and low-middle SDI regions, CVD mortality peaked in the high-middle group, and neoplasm mortality declined steadily from high- to low-SDI regions. CRD mortality was highest in low-middle SDI regions. DALYs followed a mixed pattern: diabetes and neoplasms were most burdensome in high-SDI regions, while CVDs peaked in the high-middle group and CRDs in the low-middle group.

Regional comparisons **(S1, S1C, Fig S1B in** S1 File**)** revealed distinct geographic disparities. Incidence was highest in the Americas for diabetes, CRDs, and neoplasms, while Europe led for CVDs. Mortality patterns mirrored these findings, with Europe highest for CVDs and neoplasms, South-East Asia for CRDs, and the Americas for diabetes. DALYs followed similar trends, with Europe highest for CVDs and neoplasms, South-East Asia for CRDs, and Africa generally lowest across all metrics, except for CRD incidence, which remained relatively high.

## Country-based analysis (2021)

Across 204 countries, as classified by the United Nations, we evaluated country-level variations in the burden of four major NCDs, cardiovascular diseases (CVDs), neoplasms, chronic respiratory diseases (CRDs), and diabetes mellitus, examining incidence, mortality, and disability-adjusted life years (DALYs). Estimates are illustrated in Fig 2, with additional details provided in **S2, S2A–2D in** S1 File.

## Cardiovascular diseases

Cardiovascular incidence was highest in Thailand (838 per 100,000) and other Southeast Asian nations, while lower rates were observed in the Philippines (450 per 100,000). Mortality was greatest in Eastern Europe, particularly Bulgaria (1,114

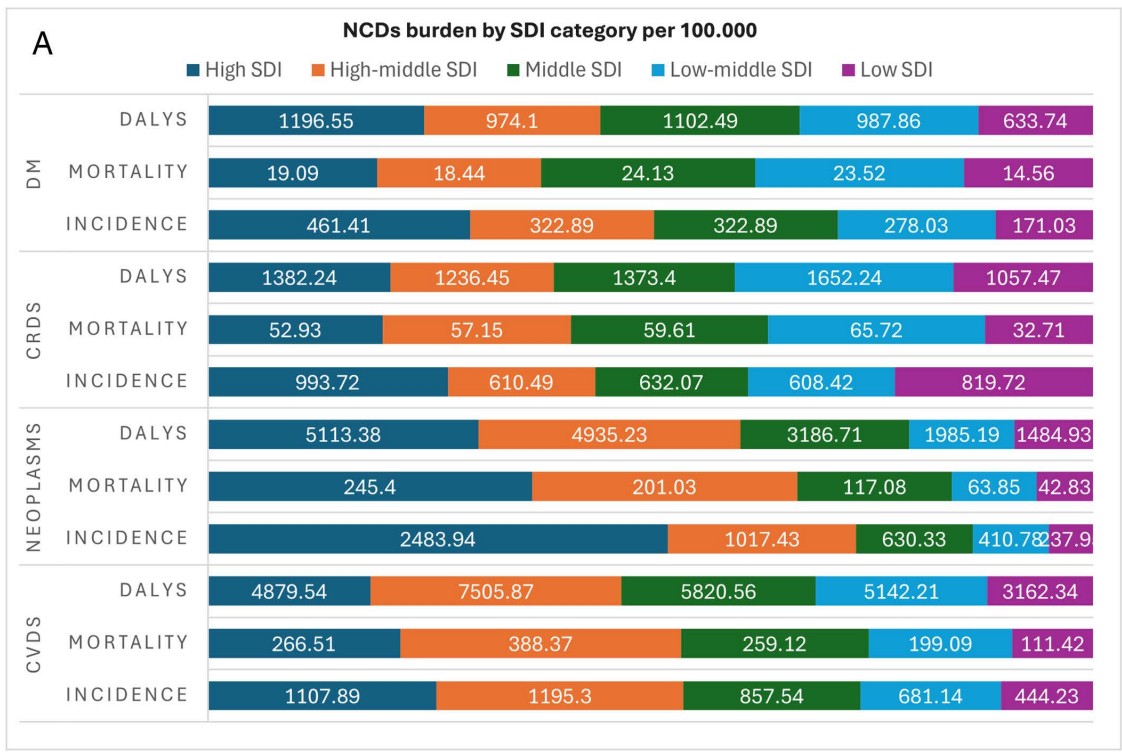

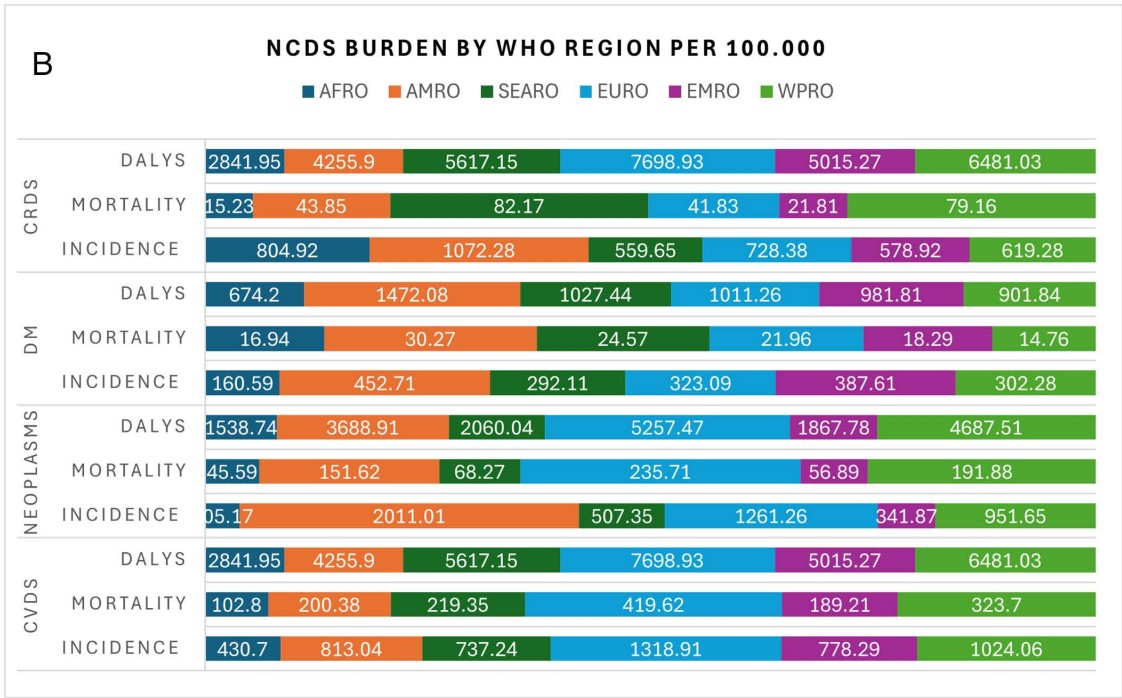

**Fig 1. Global burden of NCDs (A) SDI category and (B) WHO regions in 2021.**

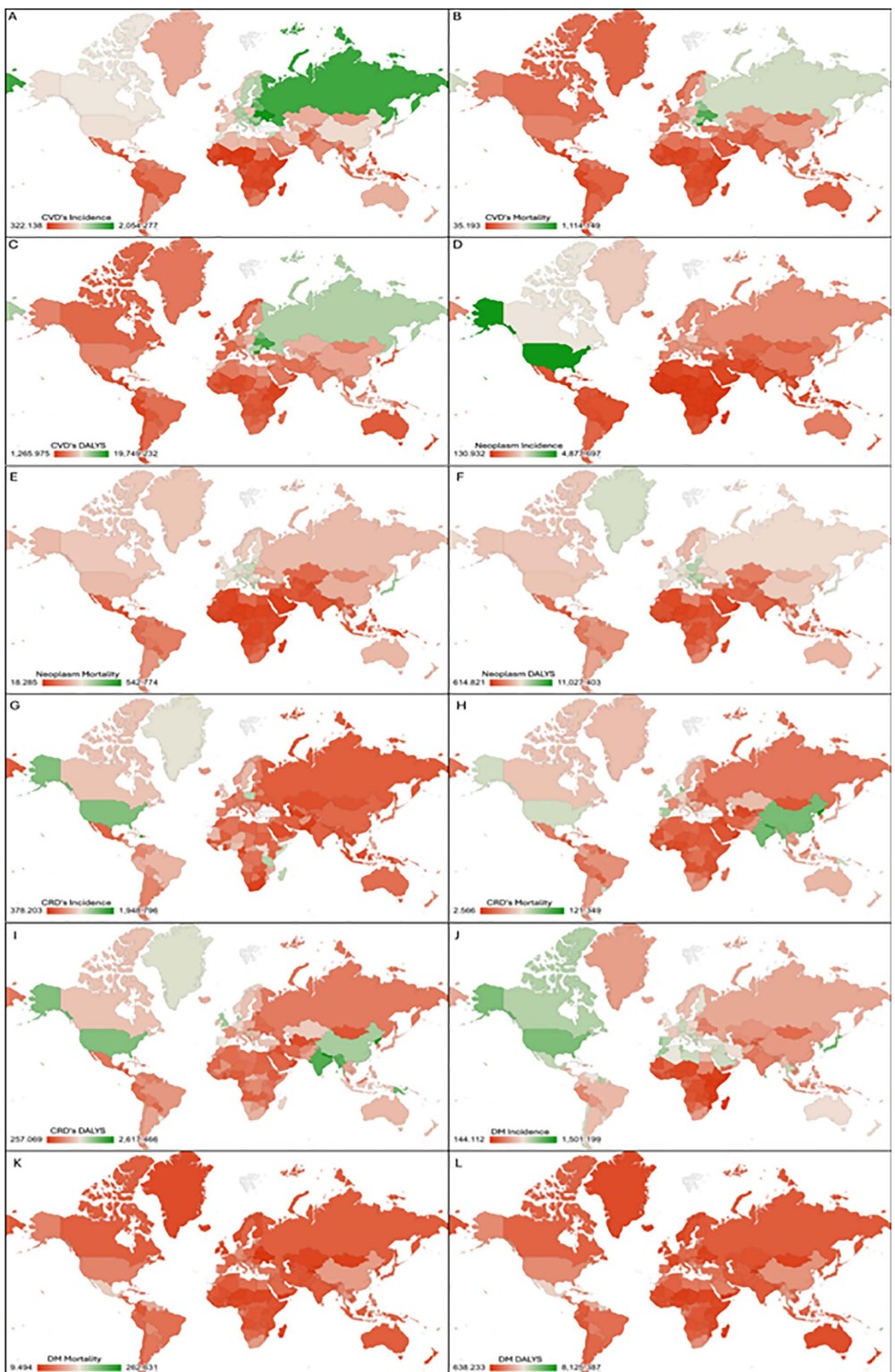

**Fig 2.  Country-based Incidence, Mortality, and DALYS rate (per 100,000 population) (2021) CVDs.** (**A, B** and **C**), Neoplasm (**D, E**, and **F**), CRDs (**G, H** and **I**), and DM (**J, K** and **L**).

per 100,000) and Ukraine (954), but much lower in countries such as Qatar (35) and the UAE (53). DALY rates followed a similar pattern, with Bulgaria and Ukraine showing the highest burdens, compared with markedly lower values in Qatar (1,266) and Ethiopia (1,812) **(S2, S2A in S1 File)**.

### Neoplasms

Incidence was highest in the U.S. (4,878 per 100,000) and other high-income countries, while lowest rates were reported in Somalia (131) and Burundi (139). Mortality peaked in Monaco (543 per 100,000) and Japan (363) but was lowest in Oman (18) and Niger (25). DALY rates showed the same gradient, with Monaco (11,027 per 100,000) and Bulgaria (7,961) at the top, compared to Oman (615) and Niger (877) at the bottom (**S2, S2B in S1 File**).

### Chronic respiratory diseases

Incidence was greatest in Haiti (1,949 per 100,000) and the U.S. (1,545), while the lowest rates were recorded in Lesotho (378) and Maldives (385). Mortality ranged from very high in DPR Korea (121) and Nepal (110) to very low in Qatar (3) and Kuwait (3). DALY rates showed similar contrasts, from DPR Korea (2,617) and Nepal (2,608) to Qatar (257) and Kuwait (284), (**S2, S2C in S1 File**).

### Diabetes mellitus

Incidence was highest in the U.S. Virgin Islands (1,471 per 100,000), Palau, and Puerto Rico, with the lowest rates in Somalia (153) and Kenya (155). Mortality was greatest in Mauritius (263) and Fiji (231), while the lowest levels were seen in Ukraine (9) and Tajikistan (10). DALY rates peaked in Micronesia (4,465) and Northern Mariana Islands (4,098), compared with much lower burdens in Sao Tome and Principe (1,509) and Cabo Verde (1,664) (**S2, S2D in S1 File**).

### Risk factors analysis

Globally, *metabolic risk factors* were the primary drivers of NCDs DALYs, Fi 3A, with high systolic blood pressure contributing 131·49 deaths, **Fig 3B**, and 2,718·40 DALYs per 100,000 for CVDs. *High body-mass index* (BMI) and high *fasting plasma glucose* also had substantial impacts, accounting for 24·13 deaths and 575·65 DALYs (BMI) and 28·04 deaths and 544·28 DALYs (glucose) for CVD, 4·52 deaths and 112·71 DALYs (BMI) and 4·16 deaths and 89·35 DALYs (glucose) for neoplasms, and 9·17 deaths and 498·13 DALYs (BMI) and 20·99 deaths and 1,000·20 DALYs (glucose) for diabetes. *Behavioural risk factors* amplified the NCD burden, with tobacco use linked to 36·01 deaths and 933·95 DALYs for CVD and 26·74 deaths and 641·25 DALYs for neoplasms, and *dietary risks* associated with 73·93 deaths and 1,700·34 DALYs for CVD and 8·49 deaths and 207·87 DALYs for neoplasms **Fig 3C and 3D**)· *Environmental risk factors*, particularly air pollution, were notable, with 56·80 deaths and 1,262·62 DALYs for CVD and 22·58 deaths and 477·64 DALYs for chronic respiratory diseases, **S3 Table S in S1 File**.

### Metabolic risk factors by WHO regions and SD categories

WHO regions: CVDs linked to high systolic blood pressure (SBP) is the leading cause of deaths, across WHO regions, with the highest rates in Europe (223·7 deaths and 3,912·2 DALYs per 100,000), **S1 Fig** and **S2 Fig in S1 File**, respectively. BMI and elevated FPG also significantly contribute to diabetes and neoplasm burdens, particularly in the Americas and Europe, **S3A in S1 File**.SDI category: CVDs linked to SBP show the highest death, **S3 Fig in S1 File** and DALY, S **Fig 4**, burdens across all SDI levels, peaking in high-middle SDI (215·85 deaths, 3,999·29 DALYs per 100,000). As SDI increases, CVD burdens from LDL, BMI, and FPG rise, with diabetes and neoplasms tied to BMI and FPG also increasing, diabetes + FPG DALYs reach 1,196·51 in high SDI. In contrast, CRD's linked to BMI show low, variable DALYs, peaking at 70·85 in high SDI despite lower mortality. **S3B in S1 File**.

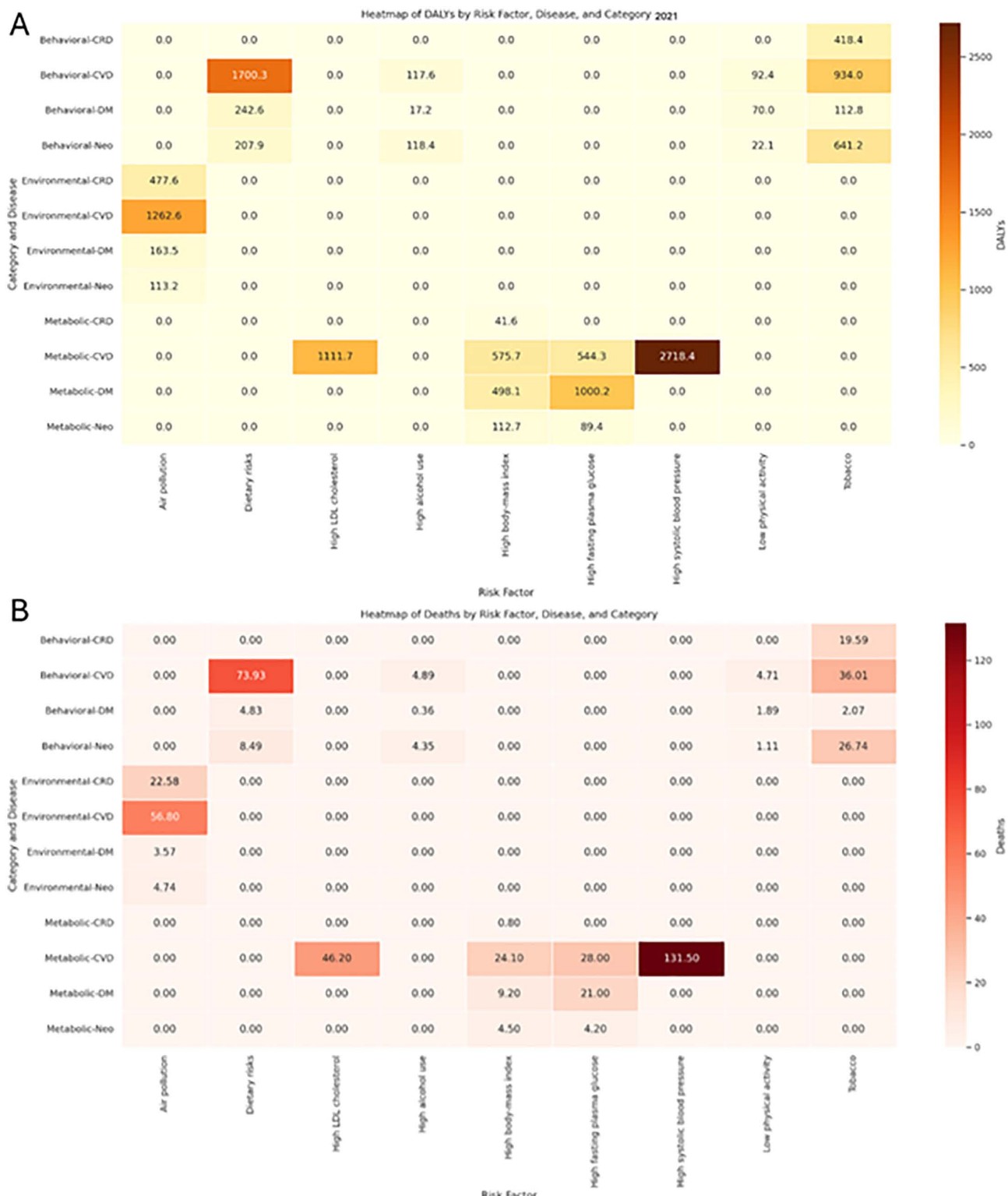

**Fig 3. Metabolic, Behavioural, and Environmental risk factors DALYs and Mortality 2021.**

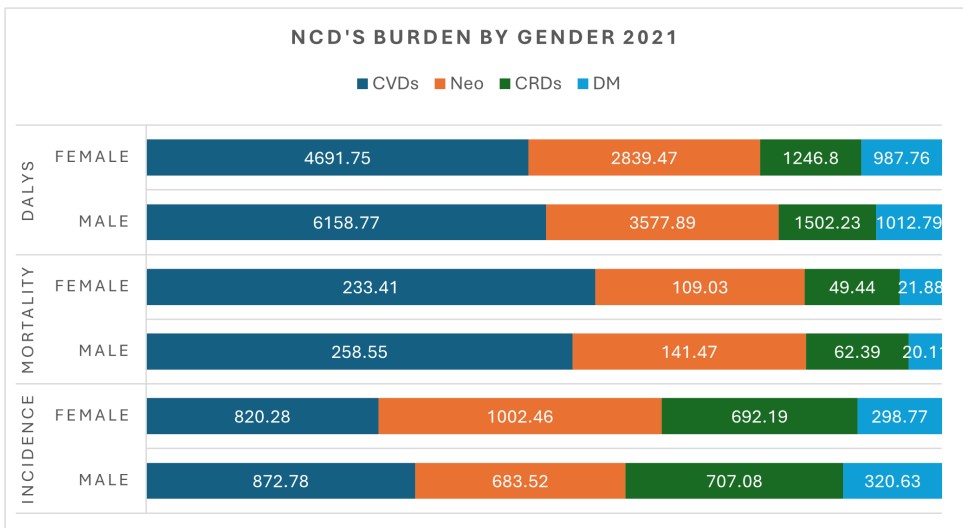

**Fig 4. Gender analysis of NCDs burden 2021.**

## Behavioural risk factors by WHO regions and SDI categories

WHO regions: The European Region records the most CVD's deaths rates from dietary risks (122·62) and alcohol (12·42), while the Western Pacific Region leads in tobacco-related deaths for CVD's (61·16) and neoplasms (54·43. The African Region consistently shows the lowest rates, such as 7·29 for CV's deaths from tobacco, **S5 Fig in S1 File** The European Region shows the highest burdens from tobacco-related cardiovascular (1,238·08 per 100,000) and neoplasm DALYs (1,137·80), as well as dietary risks (2,286·43). South-East Asia and the Eastern Mediterranean Region also report high diet-related cardiovascular DALYs (2,021·88 and 1,746·01, respectively). In contrast, the African Region has the lowest behavioural DALY burdens across most categories, particularly for alcohol and low physical activity, **S6 Fig**, **S3C in S1 File**.

SDI category: CVDs deaths from tobacco increase from 10·16 per 100,000 in low SDI to 62·62 in high-middle, then drop to 29·51 in high SDI, **S7 Fig in S1 File**. Dietary risks also peak in high-middle SDI with 112·50 CVD deaths and 2,257·85 DALYs, Fig S8 in **S1 File**. Tobacco-related neoplasm deaths rise steadily, reaching 49·87 in high SDI. CRD's and diabetes show stable burdens, with slight peaks in middle and high-middle SDI, **S3D in S1 File**.

## Environmental risk factors by WHO regions and SDI category

South-East Asia reports the highest death rates for chronic respiratory diseases (43·93) and cardiovascular diseases (69·31), **S9 Fig in S1 File**, with DALYs of 968·46 and 1716·58, respectively, **S10 Fig in S1 File**. The Western Pacific Region shows the highest neoplasm deaths (12·23). The African Region and Region of the Americas have the lowest rates, with the Americas recording the fewest cardiovascular disease deaths (16·46) and DALYs (345·25). **S3E in S1 File**.

Air pollution's health impact peaks in middle and high-middle SDI, with cardiovascular disease deaths highest in high-middle SDI (70·74), **S11 Fig in S1 File** and DALYs in middle SDI (1462·80), **S12 Fig in S1 File**. Chronic respiratory disease deaths (35·45) and DALYs (786·60) peak in low-middle SDI, while neoplasm deaths rise with SDI, reaching 9·55 in high-middle SDI. Diabetes mellitus death rates remain stable, peaking slightly in Low-middle SDI (4·39). **S3F in S1 File**.

## Gender differences in NCD burden 2021

Cardiovascular diseases were the leading cause of mortality and disability among the four major NCDs, with higher global rates in males for both deaths (258.6 vs 233.4 per 100,000) and DALYs (6158.8 vs 4691.8 per 100,000). Neoplasms

ranked second, with higher male death (141.5 vs 109.0) and DALY rates (3577.9 vs 2839.5), though incidence was notably higher in females (1002.5 vs 683.5). Chronic respiratory diseases also showed higher rates in males across deaths (62.4 vs 49.4), DALYs (1502.2 vs 1246.8), and incidence (707.1 vs 692.2). Diabetes mellitus displayed a mixed pattern: females had a slightly higher death rate (21.9 vs 20.1), while males showed higher DALYs (1012.8 vs 987.8) and incidence (320.6 vs 298.8). (**S4 in** **S1 File**, **Fig 4**).

Across WHO regions, males generally exhibited higher death and DALY rates than females for most NCDs. The main exception was in the European Region, where female cardiovascular mortality exceeded that of males (442.5 vs 395.5). Incidence patterns also varied, with females showing higher neoplasm incidence in the Region of the Americas (2176.2 vs 1839.4). **(S4A, S13 Fig in** **S1 File**). By SDI category, males experienced higher death and DALY rates across most NCDs and SDI levels. However, for neoplasms in low-SDI regions, females had higher mortality (45.2 vs 40.5) and DALYs (1617.3 vs 1352.7). Incidence data showed females consistently having higher neoplasm rates across all SDI groups, while males had higher diabetes incidence in high-SDI regions (519.6 vs 403.5). **(S4B, S14 Fig in** **S1 File**).

### Analyses of CVD's, Neoplasms, CRD's, and DM trends from 2000 to 2021

Between 2000 and 2021, global trends in non-communicable diseases (NCDs) revealed divergent patterns across CVDs, neoplasms, CRDs, and DM, as measured by age-standardized incidence, mortality, and DALY rates per 100,000 population, **Fig 5**. The figure illustrates that incidence and mortality related to diabetes mellitus (DM) increased steadily, CVDs showed a gradual rise in incidence with relatively stable mortality and slightly declining DALYs, neoplasms demonstrated consistent upward trends across incidence and mortality, while chronic respiratory diseases (CRDs) remained broadly stable or showed slight declines. Quantitative estimates of these changes, expressed as average annual percentage changes (AAPCs), are reported in **S5**: CVDs (mortality: 0.21%; incidence: 1.01%; DALYs: –0.11%), neoplasms (mortality: 0.53%; incidence: 1.01%; DALYs: 0.07%), CRDs (mortality: –0.02%; incidence: –0.36%; DALYs: –0.34%), and DM (mortality: 1.72%; incidence: 2.41%; DALYs: 2.22%).

### Global burden projections 2050

All NCDs are projected to cause approximately 75·5 million deaths and 2·44 billion DALYs globally. Among these, CVDs are expected to contribute the largest share of mortality, accounting for 86·1% of all NCDs deaths, and 49·0% of total DALYs. Neoplasms are projected to account for 24·4% of NCDs deaths and 15·9% of DALYs, while CRDs are estimated to contribute 11·1% of deaths and 6·7% of DALYs. DM is projected to make up 3·9% of NCDs deaths and 6·5% of DALYs, **S6**. The data below include age-standardized estimates of mortality and DALYs for major NCD's across countries and regions.

### Cardiovascular diseases

The highest mortality rates (per 100,000 population) are seen in Eastern Europe and Central Asia (Ukraine 1,203·77; Bulgaria 1,051·62), while the lowest are in sub-Saharan Africa (Uganda 57·89; Mali 63·92). Western Europe and the Americas report moderate rates, such as Germany (400·95) and the U.S. (318·71), **Fig 6A**. Projected DALY rates from cardiovascular diseases in 2050 show wide global disparities from 1,653 per 100,000 in Uganda to 18,688 in Ukraine, an over 11 fold gap. Eastern Europe and Central Asia have the highest rates (Ukraine 18,688; Bulgaria 16,635), while sub-Saharan Africa reports the lowest (Uganda 1,653; Mali 1,815). High-income countries like the U·S· (5,247) and Germany (5,726) show moderate levels, **Fig 6B**. **S6A in** **S1 File**.

### Neoplasms

The highest mortality rates are observed in high-income countries such as Monaco (790·7), Croatia (590·1), and Andorra (577·3), while the lowest rates are concentrated in sub-Saharan Africa, including Niger (25·7), Chad (34·2), and Mali

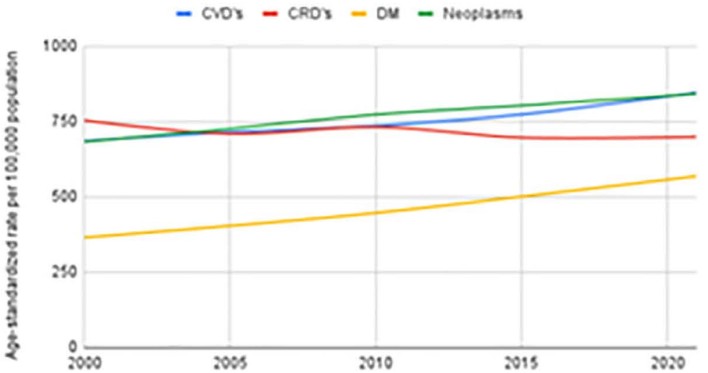

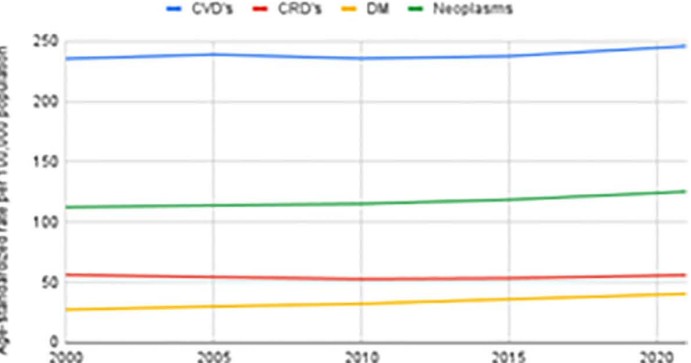

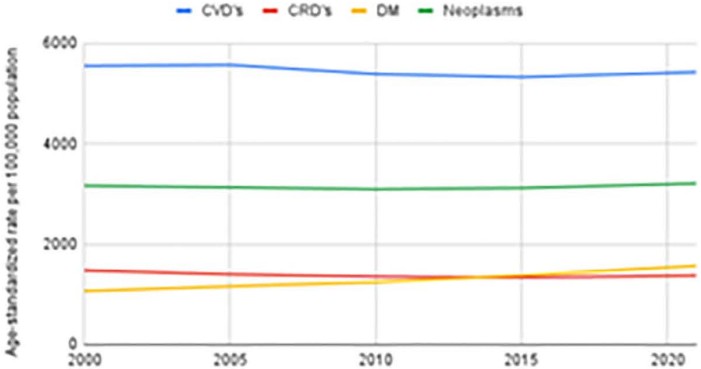

**Fig 5. Age-standardized incidence, mortality, and disability-adjusted life year (DALY) trends per 100,000 population for cardiovascular diseases (CVDs), neoplasms, chronic respiratory diseases (CRDs), and diabetes mellitus (DM), 2000–2021.** Incidence values represent rates per 100,000 population. Corresponding absolute case numbers for 2021 (e.g., CVDs = 66.8 million new cases) are reported in the results section.

(48·9), **Fig 6C**. The highest DALY burdens are concentrated in Eastern and Southern Europe, including Monaco (12,980), Croatia (10,386), and Lithuania (10,061). In contrast, the lowest DALYs are found in sub-Saharan African countries such as Niger (841), Chad (1,128), and Benin (1,449), **Fig 6D**, **S6B in S1 File**.

### Chronic respiratory diseases

The highest mortality rates are projected in East and South Asia, with DPR Korea (270·0), China (228·4), and India (145·4) among the most affected. Conversely, the lowest rates are concentrated in sub-Saharan Africa, including Burkina Faso (6·9), Chad (7·3), and Niger (8·2), **Fig 6E**. The highest DALY burdens are expected in East and South Asia, including DPR Korea (4,565), China (3,504), and India (3,025). Meanwhile, the lowest DALYs are projected in sub-Saharan Africa, with Chad (366), Burkina Faso (381), and Ethiopia (410) among the least affected, **Fig 6F**, **S6C in S1 File**.

### Diabetes mellitus

Projected DM death rates show wide global disparities, ranging from 4·8 per 100,000 in Singapore to 256·98 in Fiji. Countries with the highest rates include Fiji (256·98), Trinidad and Tobago (214·65), Mauritius (205·62), Cook Islands (193·07), and Palau (185·07). In contrast, Singapore (4·8), Ukraine (5·7), Mongolia (6·72), and Japan (7·77) report rates below

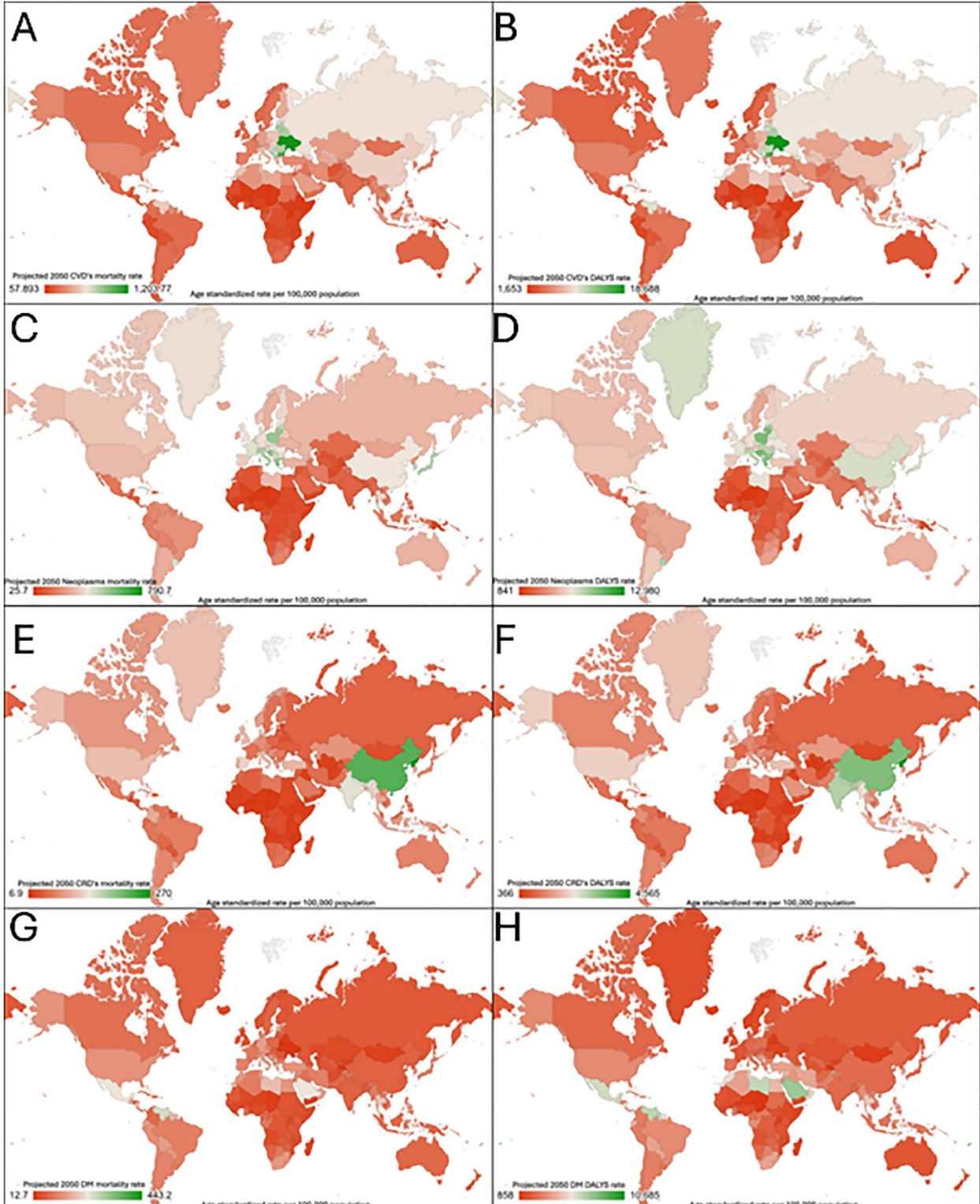

**Fig 6. 2050 Mortality and DALYS rates (per 100,000 population) projection for CVD's (A and B), Neoplasms (C and D), CRDs (E and F), and DM (G and H), respectively.**

10, suggesting effective control or lower prevalence, **Fig 6G**. DALY rates due to diabetes in 2050 also vary widely, from 496·96 in Chad to 7,917·79 in Fiji, a more than 15-fold difference. Pacific and Caribbean nations, Fiji (7,917·79), Trinidad and Tobago (7,481·58), and Cook Islands (6,548·34), show the highest burden, while African countries like Chad (496·96), Niger (574·05), and Burkina Faso (625·94) have the lowest. European countries such as Denmark (917·54) and Sweden (945·49) report relatively low DALYs. Meanwhile, China (1,698·55) and India (1,693·06) show moderate rates, **Fig 6H**, **SS6D in S1 File**.

## Discussion

The NCD landscape is marked by a striking divergence in disease patterns. Cardiovascular diseases remain the leading cause of NCD mortality, projected to account for 86.1% of deaths by 2050, driven by persistent metabolic risks like hypertension [11]. Neoplasms, fuelled by tobacco use and aging populations, show steady increases in incidence, with high-income regions like the Americas reporting rates as high as 4,877.70 per 100,000 in the U.S. [9]. Diabetes mellitus, however, is the wildcard, with incidence and DALYs surging globally due to the obesity epidemic, projected to disproportionately burden regions like the Pacific islands [12]. Chronic respiratory diseases, while relatively stable, continue to reflect environmental challenges, such as air pollution in South-East Asia [13]. Our estimates for mortality, incidence, and DALYs in 2021 are broadly consistent with those reported by Li J et al [14], reflecting the robustness of Global Burden of Disease (GBD)-based projections.

This heterogeneity demands a shift from a one-size-fits-all approach to precision-targeted prevention. For CVDs, sustained efforts in hypertension management and lipid control are critical, with high systolic blood pressure linked to 131.49 deaths per 100,000 globally [15]. Diabetes requires aggressive action on diet and physical activity, addressing the "global syndemic" of obesity and metabolic dysregulation [16]. Neoplasms call for enhanced screening and risk factor reduction [17], while CRDs necessitate clean air policies alongside smoking cessation. The WHO's "best buys" (cost-effective interventions like tobacco taxation and salt reduction) provide a foundation, but their adaptation must reflect each NCD's unique trajectory [18].

The NCD burden is deeply inequitable, shaped by socioeconomic and geographic fault lines. High Socio-Demographic Index (SDI) regions face elevated DM and neoplasm rates due to aging populations and lifestyle risks, while low SDI settings suffer higher CRD mortality from environmental exposures [19]. Middle SDI countries, in epidemiological transition, struggle with rising DM amid underfunded health systems. Regionally, Europe leads in CVD mortality (e.g., Bulgaria's 1,114.15 per 100,000), the Americas in neoplasm incidence, and South-East Asia in CRD deaths [2].

Country-level disparities are stark: Qatar's CVD mortality (35.19 per 100,000) contrasts sharply with Bulgaria's, while Somalia's neoplasm incidence (130.93) pales beside the U.S.'s. Gender further complicates the picture: males globally exhibit higher CVD and neoplasm mortality, yet females in Europe show an unexpected rise in CVD deaths, possibly tied to underrecognized risks like postmenopausal hypertension [20]. These patterns reflect systemic inequities in healthcare access, prevention, and social determinants such as urbanization, education, and income, underscoring the need for an equity-driven response.

NCD risk factors are not just drivers of disease but opportunities for policy innovation. Metabolic risks, particularly hypertension, dominate CVDs, while behavioural risks, tobacco, poor diet, and physical inactivity, fuel neoplasms and DM [21]. Air pollution, a growing concern, exacerbates CRDs, contributing to 7 million deaths annually [13]. These risks are actionable: salt reduction campaigns could avert millions of CVD deaths, tobacco taxes could curb neoplasms, and clean air initiatives could reduce CRD burdens [16].

The Pacific islands, facing a DM crisis, could pioneer community-based prevention, blending traditional diets with modern health education [12]. Europe's female CVD anomaly highlights the need for sex-specific interventions, challenging the male-centric focus of past efforts [22]. Multisectoral action, spanning health, environment, and economic policy, is essential to turn risk factors into levers for equity.

The variation we observed in the burden of major NCDs can be explained by differences in demographics, exposures, and health system performance, each carrying clear policy implications. For instance, the very high cardiovascular mortality seen in Eastern Europe is closely linked to poor blood pressure control, high alcohol consumption, and weaker health systems [23]. In contrast, the sharp rise in diabetes, particularly in Pacific Island nations, reflects the global obesity epidemic and rapid shifts toward unhealthy diets [24]. The steady increase in cancer incidence is largely driven by aging populations and continued exposure to risk factors such as tobacco and alcohol [25], while the ongoing burden of chronic respiratory diseases in Asia highlights the role of air pollution and smoking [26]. These findings point to the need for context-specific prevention strategies: stronger hypertension and lipid management programs, sugar and salt reduction policies, tobacco and alcohol taxation, clean air initiatives, and more robust primary care.

Additionally, an important and novel aspect of our findings is the variation in NCD burden by sex. Overall, men experienced higher mortality and DALY rates for CVDs and neoplasms, consistent with previous evidence linking male behavioural risk patterns, such as higher tobacco and alcohol use, to excess burden [27]. However, our analysis also showed an unexpected rise in CVD mortality among women in Europe, which may be related to postmenopausal hypertension and under-recognition of female-specific cardiovascular risk factors [27]. In contrast, neoplasm incidence was consistently higher among women across SDI categories, reflecting both biological differences and greater utilization of screening services [25]. These disparities underscore the need for sex-specific prevention and management strategies, aligning with recent calls to integrate gender-sensitive approaches into NCD policy and research [28].

The 2050 projections are a clarion call: without action, NCDs will overwhelm health systems, deepened by post-COVID-19 vulnerabilities like delayed care and increased metabolic risks [14]. Scaling up primary care, integrating NCD services, and leveraging digital health are critical to bridge access gaps. The WHO's 2023–2030 NCD Roadmap offers a blueprint, but its success depends on political commitment and investment in the most vulnerable, rural populations, low-income regions, and women facing unique risks.

The NCD crisis tests our commitment to global health equity. It exposes fractures between rich and poor, urban and rural, male and female and demands a future where health is a universal right. From Bulgaria's CVD burden to Fiji's DM challenge, the stakes are clear: without bold action, NCDs will derail SDG 3.4 and widen disparities. This is a scientific, economic, and moral imperative, calling for innovation, solidarity, and a relentless focus on those left behind.

The NCD landscape is marked by a striking divergence in disease patterns. Cardiovascular diseases remain the leading cause of NCD mortality, projected to account for 86.1% of deaths by 2050, driven by persistent metabolic risks like hypertension [11]. Neoplasms, fuelled by tobacco use and aging populations, show steady increases in incidence, with high-income regions like the Americas reporting rates as high as 4,877.70 per 100,000 in the U.S. [9]. Diabetes mellitus, however, is the wildcard, with incidence and DALYs surging globally due to the obesity epidemic, projected to disproportionately burden regions like the Pacific islands [12]. Chronic respiratory diseases, while relatively stable, continue to reflect environmental challenges, such as air pollution in South-East Asia [13].

This heterogeneity demands a shift from a one-size-fits-all approach to precision-targeted prevention. For CVDs, sustained efforts in hypertension management and lipid control are critical, with high systolic blood pressure linked to 131.49 deaths per 100,000 globally [15]. Diabetes requires aggressive action on diet and physical activity, addressing the "global syndemic" of obesity and metabolic dysregulation [16]. Neoplasms call for enhanced screening and risk factor reduction, while CRDs necessitate clean air policies alongside smoking cessation. The WHO's "best buys" (cost-effective interventions like tobacco taxation and salt reduction) provide a foundation, but their adaptation must reflect each NCD's unique trajectory [18].

The NCD burden is deeply inequitable, shaped by socioeconomic and geographic fault lines. High Socio-Demographic Index (SDI) regions face elevated DM and neoplasm rates due to aging populations and lifestyle risks, while low SDI settings suffer higher CRD mortality from environmental exposures [19]. Middle SDI countries, in epidemiological transition,

struggle with rising DM amid underfunded health systems. Regionally, Europe leads in CVD mortality (e.g., Bulgaria's 1,114.15 per 100,000), the Americas in neoplasm incidence, and South-East Asia in CRD deaths [2].

Country-level disparities are stark: Qatar's CVD mortality (35.19 per 100,000) contrasts sharply with Bulgaria's, while Somalia's neoplasm incidence (130.93) pales beside the U.S.'s. Gender further complicates the picture: males globally exhibit higher CVD and neoplasm mortality, yet females in Europe show an unexpected rise in CVD deaths, possibly tied to underrecognized risks like postmenopausal hypertension [20]. These patterns reflect systemic inequities in healthcare access, prevention, and social determinants such as urbanization, education, and income, underscoring the need for an equity-driven response.

NCD risk factors are not just drivers of disease but opportunities for policy innovation. Metabolic risks, particularly hypertension, dominate CVDs, while behavioural risks, tobacco, poor diet, and physical inactivity, fuel neoplasms and DM [21]. Air pollution, a growing concern, exacerbates CRDs, contributing to 7 million deaths annually [13]. These risks are actionable: salt reduction campaigns could avert millions of CVD deaths, tobacco taxes could curb neoplasms, and clean air initiatives could reduce CRD burdens [16].

A key consideration in interpreting our findings is the nature of the GBD data. Estimates are generated from multiple sources, including civil registration systems, disease registries, surveys, hospital records, and verbal autopsies, and are harmonized through statistical modelling. While this allows for global comparability, it also introduces uncertainty, particularly in countries with limited data. Misclassification of causes of death, uneven data quality, and reliance on modelling make independent validation difficult.

Our data ended in 2021, so the full impact of the COVID-19 pandemic is not captured. Projections to 2050 are based on current trends and may not reflect future advances in prevention, treatment, or policy. The use of individual risk factors such as hypertension and tobacco may oversimplify the overlap of multiple risks, and subgroup disparities beyond sex (such as age or geography) were not examined. Finally, average annual percentage change (AAPC) was calculated using aggregated counts, which may reflect population growth and aging rather than true shifts in disease risk, though age-standardized rates were included to mitigate this.

## Conclusion

The rising global burden of NCDs, especially cardiovascular diseases and diabetes, calls for urgent, equity-focused prevention and stronger health systems. Regional, gender, and socioeconomic disparities highlight the need to tackle systemic inequities to meet Sustainable Development Goal 3.4. Despite data and projection challenges, this study offers a vital foundation for advancing WHO's NCD Roadmap and fostering a healthier future by 2050.

## Supporting information

**S1 File. Supplementary files.**
(PDF)

## Author contributions

**Conceptualization:** Omar Freihat, Arpad Kovacs.

**Data curation:** Omar Freihat.

**Formal analysis:** Omar Freihat.

**Validation:** Dvid Sipos.

**Writing – original draft:** Dvid Sipos.

**Writing – review & editing:** Maria Aamir, Arpad Kovacs.

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
