## [Decision Letter · Decision Letter 0]

29 Aug 2025

Dear Dr. Freihat,

Thank you for submitting your manuscript to PLOS ONE. After careful consideration, we feel that it has merit but does not fully meet PLOS ONE’s publication criteria as it currently stands. Therefore, we invite you to submit a revised version of the manuscript that addresses the points raised during the review process.

We look forward to receiving your revised manuscript.

Kind regards,

Anselm J. M. Hennis, MBBS, MSc, PhD, FRCP

Academic Editor

PLOS ONE

**Journal Requirements:**

1. When submitting your revision, we need you to address these additional requirements. Please ensure that your manuscript meets PLOS ONE's style requirements, including those for file naming. The PLOS ONE style templates can be found at https://journals.plos.org/plosone/s/file?id=wjVg/PLOSOne_formatting_sample_main_body.pdf and https://journals.plos.org/plosone/s/file?id=ba62/PLOSOne_formatting_sample_title_authors_affiliations.pdf 2. Thank you for uploading your study's underlying data set. Unfortunately, the repository you have noted in your Data Availability statement does not qualify as an acceptable data repository according to PLOS's standards. At this time, please upload the minimal data set necessary to replicate your study's findings to a stable, public repository (such as figshare or Dryad) and provide us with the relevant URLs, DOIs, or accession numbers that may be used to access these data. For a list of recommended repositories and additional information on PLOS standards for data deposition, please see https://journals.plos.org/plosone/s/recommended-repositories. 3. We note that Figures 3 and 6 in your submission contain map images which may be copyrighted. All PLOS content is published under the Creative Commons Attribution License (CC BY 4.0), which means that the manuscript, images, and Supporting Information files will be freely available online, and any third party is permitted to access, download, copy, distribute, and use these materials in any way, even commercially, with proper attribution. For these reasons, we cannot publish previously copyrighted maps or satellite images created using proprietary data, such as Google software (Google Maps, Street View, and Earth). For more information, see our copyright guidelines: http://journals.plos.org/plosone/s/licenses-and-copyright. We require you to either present written permission from the copyright holder to publish these figures specifically under the CC BY 4.0 license, or remove the figures from your submission: a. You may seek permission from the original copyright holder of Figures 3 and 6 to publish the content specifically under the CC BY 4.0 license.   We recommend that you contact the original copyright holder with the Content Permission Form (http://journals.plos.org/plosone/s/file?id=7c09/content-permission-form.pdf) and the following text:“I request permission for the open-access journal PLOS ONE to publish XXX under the Creative Commons Attribution License (CCAL) CC BY 4.0 (http://creativecommons.org/licenses/by/4.0/). Please be aware that this license allows unrestricted use and distribution, even commercially, by third parties. Please reply and provide explicit written permission to publish XXX under a CC BY license and complete the attached form.” Please upload the completed Content Permission Form or other proof of granted permissions as an "Other" file with your submission. In the figure caption of the copyrighted figure, please include the following text: “Reprinted from [ref] under a CC BY license, with permission from [name of publisher], original copyright [original copyright year].” b. If you are unable to obtain permission from the original copyright holder to publish these figures under the CC BY 4.0 license or if the copyright holder’s requirements are incompatible with the CC BY 4.0 license, please either i) remove the figure or ii) supply a replacement figure that complies with the CC BY 4.0 license. Please check copyright information on all replacement figures and update the figure caption with source information. If applicable, please specify in the figure caption text when a figure is similar but not identical to the original image and is therefore for illustrative purposes only.The following resources for replacing copyrighted map figures may be helpful: USGS National Map Viewer (public domain): http://viewer.nationalmap.gov/viewer/The Gateway to Astronaut Photography of Earth (public domain): http://eol.jsc.nasa.gov/sseop/clickmap/Maps at the CIA (public domain): https://www.cia.gov/library/publications/the-world-factbook/index.html and https://www.cia.gov/library/publications/cia-maps-publications/index.htmlNASA Earth Observatory (public domain): http://earthobservatory.nasa.gov/Landsat:
http://landsat.visibleearth.nasa.gov/USGS EROS (Earth Resources Observatory and Science (EROS) Center) (public domain): http://eros.usgs.gov/#Natural Earth (public domain): http://www.naturalearthdata.com/ 4. Please include captions for your Supporting Information files at the end of your manuscript, and update any in-text citations to match accordingly. Please see our Supporting Information guidelines for more information: http://journals.plos.org/plosone/s/supporting-information. 5. If the reviewer comments include a recommendation to cite specific previously published works, please review and evaluate these publications to determine whether they are relevant and should be cited. There is no requirement to cite these works unless the editor has indicated otherwise. 

**Additional Editor Comments:**

This paper describes the global burden of NCDs based on GBD data. While this has been done by others in the past, the authors make projections to 2050 which adds value. As noted by both Reviewers, there are significant concerns about the use of GBD data to construct mortality and other outcomes. The AAPC formula uses aggregated counts instead of age-standardized rates, and counts make comparability over time questionable. These intrinsic limitations with the data sources used must be addressed by the authors. Furthermore, the accuracy, comprehensiveness and readability of the information presented could be significantly improved, and the Reviewers have both provided detailed feedback to the authors. In sum, this paper requires a major revision before it can be considered for publication.

Reviewers' comments:

**Comments to the Author**

1. Is the manuscript technically sound, and do the data support the conclusions?

Reviewer #1: Partly

Reviewer #2: Partly

2. Has the statistical analysis been performed appropriately and rigorously?

Reviewer #1: Yes

Reviewer #2: Yes

3. Have the authors made all data underlying the findings in their manuscript fully available?

Reviewer #1: Yes

Reviewer #2: Yes

4. Is the manuscript presented in an intelligible fashion and written in standard English?

Reviewer #1: Yes

Reviewer #2: Yes

**Reviewer #1:**  The manuscript addresses an important topic about the growing burden of non-communicable diseases in different regions around the world. However, major revisions are recommended to clarify methodology, condense results, correct inconsistencies, spelling and grammar.

In the introduction section, there is a sentence which states:

“This paper examines the global burden of NCDs by analysing trends in incidence, mortality, and disability-adjusted life years (DALYs) across regions and income levels from 2000 to 2021, with projections extending to 2025”. Should this not read, “…projections extending to 2050” as is outlined in the Methods and Results sections?

The abstract and the methods report the data source year inconsistently – both GBD 2019 and 2021 are mentioned. This needs to be clarified.

The AAPC formula uses aggregated counts instead of age-standardised rates. Counts make comparability over time questionable. If aggregated counts are used, this mostly reflects population growth and aging and not necessarily true changes in disease risk. The methodology should be clarified, and, limitations of this method described.

The results section is comprehensive, but it is very difficult for readers to follow. I would suggest condensing all the numeric detail into more readable sentences and using tables/graphs to show disparities and trends. Exhaustive lists can be provided as supplementary data.

Figure 3 needs to be altered so that there is no confusion with the graph and the labels. Scales and annotations to the figures should not overlap with other figures/maps

Captions can provide more information about the figures.

The discussion can be improved by linking each major finding with plausible explanations and potential health policy implications.

Suggest expanding on some of the novel findings like the sex differences, which is interesting.

Spelling switches between US and UK English throughout the manuscript. This needs to be carefully checked and corrected.

Spelling: “3.5.4. Diabetes Maleates” and “3.8.4. Diabetes Maleates”

**Reviewer #2:**  Comments to the authors,

This is a well-structured manuscript that describes the burden of NCDs using GBD data. Although there are several published studies reporting similar findings with this data source, the authors have added projections to 2050, which strengthens the manuscript. I have made several comments to improve the content, below. One of the major issues when using GBD data is addressing the technical pitfalls related to the identification of original information to construct mortality and other outcomes. The manuscript makes no mention to this issue (other than briefly outlining one of the limitations in the use of SDIs) and I would recommend that this gap is carefully addressed in the discussion.

Abstract –

Background, please edit first line to clarify that NCDs, [including] CVD etc are the leading cause of death. It might be argued that NCDs are not the leading global health challenge

Introduction –

Well-structured and places the topic into context.

Page 9, paragraph 1 - I suggest editing the first paragraph to clarify that NCDs disproportionally affect the most vulnerable socio-economic communities (there appears to be a disconnection between sentences linking the socio-economic contexts [not driven by urbanization or ageing populations)]

Page 9, paragraph 3 - The Intro states that projections of estimates extend to 2025, while the abstract describe them as of 2050 – other sections of the manuscript (e.g., Methods) also refer to either, please review for consistency

Methods –

A description is required of what was included as NCDs to quantify incidence, mortality and DALYs – please provide details with levels of NCD classification i.e., level 1, 2 and 3, and include the information in a supplementary table

Page 10, paragraph 1 – Unclear why selecting 204 countries ‘ensure methodological consistency with international standard’ (please clarify); same paragraph, please add reference for DALYs

Page 10, paragraph 2 – when first introducing standardized rates, please state 100,000 ‘population per year, or 100,000 ‘’people per year. For subsequent mentions in the manuscript, 100,000 would suffice’. Please add reference to cite DALYs and SDI

Results –

The majority of results on mortality/incidence and DALYs reported for 2021 replicate those by Li et al (Int J Surgery, May 2025) – I would suggest citing them and further differentiating the reporting of findings; also please indicate in the Discussion how the two studies are complementary (where not overlapping)

Page 11, paragraph 1 – it is stated that the global incidence of NCDs was 12.4 billion new cases, and the four most common types CVD (66.8 million), cancer (66.5 million), chronic respiratory disease 55.2 million, and DM (24.4 million) represent less than 3% of all global NCD incidence. It would be helpful to report this, and to indicate what the leading incident NCDs were. Is the incidence rate correct (156,680 per 100,000 population)?

Figure 1 – it shows incidence trends for the period 200-2021, however, the figures for the incidence in 2021 does not appear to match what is described in paragraph 1 of Results. In figure 1, the incidence of CVD for 2021 is ~850 (million?) with similarly high numbers reported for the other three causes. Please clarify.

Figure 1 – it needs to indicate the unit of the incidence trends (e.g., age-standardised rates per …, or just raw numbers? ) if the info is incidence per 100,000 population, the numbers do not align with what is described in the first paragraph of results for incidence

Page 11, section 3.2. – the description provided in this paragraph does not match Figure 1, the AAPC are not appreciated in Figure 1 (they are in S Table 1 B, which his correctly cited). Figure 1 would need to have more information about what axis Y and X represent, and a brief description of the trends observable in the figure, provided

Figures 2A and 2B – please include the year for which the estimates are provided, I assume it is 2021 ?

Page 12, section 3.3. – I would suggest reordering this paragraph to make the summary of findings easier to read (please note some punctuation is also missing. Rather than repeating all the info provided in Figure 2A, I suggest complementing it in the text, and summarizing in which SDI regions incidence, mortality, and DALYs for each of the four NCDs were highest and lowest

Page 14, country-based analysis – please indicate in this paragraph a brief summary of the types of NCDs included and mention that estimates are illustrated in Figure 3 (rather than repeating ‘figure 3’ in each sub-section of NCDs)

Pages 15 and 23, I think the authors mean diabetes mellitus, not diabetes maleates ? The abstract, Methods and Intro sections describe it as the former.

Page 18, Figure 4 – please insert the year (2021) for which the reporting of metabolic and other risk factors is reported is presented in this figure (same applies for figure 5)

Discussion – the main findings are summarized and the impact of the magnitude of NCD burden is outlined. However, no discussion is given to the type of data used to determine NCD incidence or mortality, and this is a critical aspect when interpreting data from the GBD consortium. This should be described and discussed (in addition to being stated as a limitation of the study in the corresponding section, which is also not mentioned)

**Do you want your identity to be public for this peer review?** For information about this choice, including consent withdrawal, please see our Privacy Policy

Reviewer #1: No

Reviewer #2: No

---

## [Author Response · Author response to Decision Letter 1]

25 Sep 2025

Dear Editor:

Editor’s comments

Dear Editor, thank you very much for your time and efforts, below we are highlighting the changes made as er your valuable comments:

Modifications: All required were followed as per the given guidelines.

Action: the data was uploaded to a repository and the link was modified in the text, please see: https://figshare.com/account/articles/30146254?file=58030711

3. We note that Figures 3 and 6 in your submission contain map images which may be copyrighted. All PLOS content is published under the Creative Commons Attribution License (CC BY 4.0), which means that the manuscript, images, and Supporting Information files will be freely available online, and any third party is permitted to access, download, copy, distribute, and use these materials in any way, even commercially, with proper attribution. For these reasons, we cannot publish previously copyrighted maps or satellite images created using proprietary data, such as Google software (Google Maps, Street View, and Earth). For more information, see our copyright guidelines: http://journals.plos.org/plosone/s/licenses-and-copyright.

Reply: Thank you for your note regarding Figures 3 and 6. I would like to clarify that these figures were not taken from any external copyrighted source. They were generated directly by us using Google Spreadsheet (Charts) based on the study’s dataset. No copyrighted maps or satellite images (e.g., from Google Maps, Earth, or similar) were used. The underlying data used to create the figures are openly available, and the dataset links have been provided to ensure transparency and reproducibility. Therefore, the figures are original outputs of our analysis and fully compliant with the CC BY 4.0 license. Please find the links below that show the data (open access)

1. https://docs.google.com/spreadsheets/d/1CAjKFXkL6xAJQqDCS3JhcHQ4V1jT7TiZbD8fI3s6ux4/edit?usp=sharing

2. https://docs.google.com/spreadsheets/d/18jAystyitaQS9ThZBQ6DpUBTrF3H3KDeh2AjaivMM-4/edit?usp=sharing

3. https://docs.google.com/spreadsheets/d/1a0E-cNhes6aWFCxmh90ZkiWJFOMZ3eMA9PKeYrsQsoU/edit?usp=sharing

4. https://docs.google.com/spreadsheets/d/1BYZw6ctc6eK9XJDe8iBIXe5Uu8yYd0CaoCZs7oXGgZ4/edit?usp=sharing

5. https://docs.google.com/spreadsheets/d/1Gvokj169uhC0Q36ae_t5Xyf4P55QlLY8vFW4aw92JJA/edit?usp=sharing

Action: supporting information files were modified as required.

Response: Not applicable

Additional Editor Comments:

This paper describes the global burden of NCDs based on GBD data. While this has been done by others in the past, the authors make projections to 2050 which adds value. As noted by both Reviewers, there are significant concerns about the use of GBD data to construct mortality and other outcomes. The AAPC formula uses aggregated counts instead of age-standardized rates, and counts make comparability over time questionable. These intrinsic limitations with the data sources used must be addressed by the authors. Furthermore, the accuracy, comprehensiveness and readability of the information presented could be significantly improved, and the Reviewers have both provided detailed feedback to the authors. In sum, this paper requires a major revision before it can be considered for publication.

Response: All comments were addressed as requested, please track the changes on the revised version and the final manuscript.

Dear reviewer (1):

Thank you very much for your time and efforts for reviewing our paper, we appreciate your contribution to enriching our paper. As to address the important point you have raised, we kindly send the response point by point with track changes in the attached document.

In the introduction section, there is a sentence which states:

“This paper examines the global burden of NCDs by analysing trends in incidence, mortality, and disability-adjusted life years (DALYs) across regions and income levels from 2000 to 2021, with projections extending to 2025”. Should this not read, “…projections extending to 2050” as is outlined in the Methods and Results sections?

Reply: thank you for your observations, this has acknowledged and modified.

The abstract and the methods report the data source year inconsistently – both GBD 2019 and 2021 are mentioned. This needs to be clarified.

Reply: Thank you for pointing this out. The study used GBD 2021 data from IHME, while the GBD 2019 framework was referenced for disease classification and methodology. We have revised the Abstract and Methods to clarify this distinction.

The AAPC formula uses aggregated counts instead of age-standardised rates. Counts make comparability over time questionable. If aggregated counts are used, this mostly reflects population growth and aging and not necessarily true changes in disease risk. The methodology should be clarified, and, limitations of this method described.

Reply: Thank you very much for your comment, we have used the rates as shown in the formula, we mentioned raw GDB which mean (rate per 100,000) as the source but not as the calculations, the calculations were made using the formula below

AAPC = ((〖Rates in〗⁡2021/(Rates in 2000))^(1/21)-1)*100%

We further elaborate more in the methods and added a paragraph as a limitation.

The results section is comprehensive, but it is very difficult for readers to follow. I would suggest condensing all the numeric detail into more readable sentences and using tables/graphs to show disparities and trends. Exhaustive lists can be provided as supplementary data.

Reply: Thank you for the suggestion. We revised the Results to improve readability: numeric detail has been condensed into brief summary sentences (highlighting highest/lowest patterns), with disparities and trends shown in Figures 2A/2B and Figure3. Full country and regional values are moved to Supplementary Tables 1C–1D and 2A–2D. This reorganization aligns the text with the figures and avoids exhaustive lists in the main manuscript. All sections were revised and rearranged.

Figure 3 needs to be altered so that there is no confusion with the graph and the labels. Scales and annotations to the figures should not overlap with other figures/maps

Reply: the figure was replaced, and the notations were adjusted

Captions can provide more information about the figures.

Reply: captions were modified

The discussion can be improved by linking each major finding with plausible explanations and potential health policy implications

Reply: Thank you for the suggestion. We have revised the Discussion to link each major finding with plausible explanations such as demographic change, risk factor exposures, and health system performance and highlighted policy implications in line with WHO SDG 3.4.

Suggest expanding on some of the novel findings like the sex differences, which is interesting.

Reply: a paragraph was added to highlight the gender differences.

Spelling switches between US and UK English throughout the manuscript. This needs to be carefully checked and corrected.

Reply: addressed

10. Spelling: “3.5.4. Diabetes Maleates” and “3.8.4. Diabetes Maleates”

Reply: addressed

Dear reviewer (2):

Thank you very much for your time and efforts for reviewing our paper, we appreciate your contribution to enriching our paper. As to address the important point you have raised, we kindly send the response point by point with track changes in the attached document.

1. “One of the major issues when using GBD data is addressing the technical pitfalls related to the identification of original information to construct mortality and other outcomes. The manuscript makes no mention to this issue (other than briefly outlining one of the limitations in the use of SDIs) and I would recommend that this gap is carefully addressed in the discussion.”

Reply: We thank the reviewer for this important observation. We have revised the discussion to explicitly acknowledge the technical pitfalls of GBD data, particularly the difficulty in identifying and validating the original source information used to construct mortality and outcome estimates. Text added to the discussion (last two paragraphs before the conclusion, page 32 track changes document): “A key methodological limitation of GBD data is the difficulty in tracing the original source information used to construct mortality and outcome estimates. Because GBD integrates diverse datasets through complex modeling, the transparency and reproducibility of the underlying inputs remain constrained. This limits independent validation and may obscure the extent to which local data are represented, an issue that should be considered when interpreting GBD-derived outcomes.”

2. Abstract –

Background, please edit first line to clarify that NCDs, [including] CVD etc are the leading cause of death. It might be argued that NCDs are not the leading global health challenge

Modifications: we acknowledge the importance of this observation, we have modified the text to the following:

“Non-communicable diseases (NCDs), such as cardiovascular diseases (CVDs), cancers, chronic respiratory diseases (CRDs), and diabetes mellitus (DM), are the leading cause of mortality worldwide and represent a major barrier to achieving Sustainable Development Goal (SDG) 3.4, which targets a one-third reduction in premature mortality by 2030. This study quantifies the current and projected burden of these NCDs, ealuates disparities, and explores prevention opportunities..”

3. Introduction –

Well-structured and places the topic into context.

Page 9, paragraph 1 - I suggest editing the first paragraph to clarify that NCDs disproportionally affect the most vulnerable socio-economic communities (there appears to be a disconnection between sentences linking the socio-economic contexts [not driven by urbanization or ageing populations)]

Page 9, paragraph 3 - The Intro states that projections of estimates extend to 2025, while the abstract describe them as of 2050 – other sections of the manuscript (e.g., Methods) also refer to either, please review for consistency.

Modified text:

Non-communicable diseases (NCDs), such as cardiovascular diseases, cancers, chronic respiratory conditions, and diabetes, are the predominant cause of death globally, accounting for approximately 74% of all mortality, as reported by the World Health Organization (1). This escalating burden, intensifying over recent decades, places significant pressure on health systems worldwide, particularly in low- and middle-income countries (LMICs), where limited resources and infrastructure exacerbate challenges (2). NCDs disproportionately affect the most vulnerable socioeconomic communities, including those in LMICs, due to factors such as poverty, limited healthcare access, and social inequities, rather than solely rapid urbanization or aging populations (3). In LMICs, this reflects an epidemiological transition from infectious diseases to chronic conditions, compounding existing health challenges (4,5).

Page 9, paragraph 3 - The Intro states that projections of estimates extend to 2025, while the abstract describe them as of 2050 – other sections of the manuscript (e.g., Methods) also refer to either, please review for consistency.

Modifications: 2025 modified to 2050

4. Methods –

A description is required of what was included as NCDs to quantify incidence, mortality and DALYs – please provide details with levels of NCD classification i.e., level 1, 2 and 3, and include the information in a supplementary table

Modifications: classification table was added to supplementary data, the modified text below replaced the original

“We conducted a secondary analysis of the global burden of non-communicable diseases (NCDs), specifically cardiovascular diseases (CVD), neoplasms, diabetes mellitus (DM), and chronic respiratory diseases (CRDs), from 2000 to 2021, with projections to 2050. These NCDs were classified using the Global Burden of Disease (GBD) Study 2019 framework, which organizes diseases into three levels: Level 1 includes all NCDs as a broad group, and Level 2 includes major categories like CVD, neoplasms, DM, and CRDs. Our study focused on these Level 2 categories without analyzing specific Level 3 subcategories (e.g., ischemic heart disease or lung cancer). Supplementary Table S1 lists the NCDs included in our analysis and their GBD classification levels. Data were sourced from the GBD 2019, published by the Institute for Health Metrics and Evaluation (IHME) (9), covering 204 countries and territories. This comprehensive country selection aligns with the GBD 2019’s standardized framework, which uses consistent data collection and reporting methods across nations to ensure comparability with global health metrics standards. The GBD 2019 provides estimates of incidence, mortality, and disability-adjusted life years (DALYs) (9), a metric combining years of life lost and years lived with disability, derived from integrated health data, along with risk factor contributions and 2050 projections based on historical trends.”

Page 10, paragraph 1 – Unclear why selecting 204 countries ‘ensure methodological consistency with international standard’ (please clarify); same paragraph, please add reference for DALYs

We selected 204 countries because this classification aligns with the Global Burden of Disease (GBD) study framework, which is widely used in global health research and adopted by major international agencies (e.g., WHO, UN). Using the same country set ensures comparability and methodological consistency with internationally recognized standards, enabling our results to be directly compared with other global and regional analyses. This approach avoids selective inclusion or exclusion of countries that could bias burden estimates and maintains transparency and reproducibility of our methodology. Reference for DALYS was added.

5. Page 10, paragraph 2 – when first introducing standardized rates, please state 100,000 ‘population per year, or 100,000 ‘’people per year. For subsequent mentions in the manuscript, 100,000 would suffice’. Please add reference to cite DALYs and SDI

Modified text (methods section):

We reported age-standardized incidence rates (new cases per 100,000 population per year), mortality rates (deaths per 100,000 population per year), and disability-adjusted life years (DALYs; per 100,000 population per year), which combine years of life lost and years lived with disability, for cardiovascular diseases (CVD), neoplasms, diabetes mellitus (DM), and chronic respiratory diseases (CRDs). Outcomes were evaluated globally, by Socio-Demographic Index (SDI; Low, Low-Middle, Middle, High-Middle, High), WHO region (African, Americas, Eastern Mediterranean, European, South-East Asia, Western Pacific), sex, and country, with highest and lowest rates highlighted where applicable.

6. Results –

The majority of results on mortality/incidence and DALYs reported for 2021 replicate those by Li et al (Int J Surgery, May 2025) – I would suggest citing them and further differentiating the re

---

## [Decision Letter · Decision Letter 1]

20 Oct 2025

Global Burden and Future Projections of Non-Communicable Diseases (2000–2050): Progress Toward SDG 3.4 and Disparities Across Regions and Risk Factors

PONE-D-25-26588R1

Dear Dr. Freihat,

We’re pleased to inform you that your manuscript has been judged scientifically suitable for publication and will be formally accepted for publication once it meets all outstanding technical requirements.

Kind regards,

Anselm J. M. Hennis, MBBS, MSc, PhD, FRCP

Academic Editor

PLOS ONE

Additional Editor Comments (optional):

The Authors have comprehensively addressed the areas of concern, and this manuscript is suitable for publication.

Reviewers' comments:

Reviewer's Responses to Questions

**Comments to the Author**

Reviewer #1: All comments have been addressed

Reviewer #2: All comments have been addressed

2. Is the manuscript technically sound, and do the data support the conclusions?

Reviewer #1: Yes

Reviewer #2: Yes

3. Has the statistical analysis been performed appropriately and rigorously?

Reviewer #1: Yes

Reviewer #2: Yes

4. Have the authors made all data underlying the findings in their manuscript fully available?

Reviewer #1: Yes

Reviewer #2: Yes

5. Is the manuscript presented in an intelligible fashion and written in standard English?

Reviewer #1: Yes

Reviewer #2: Yes

Reviewer #1: (No Response)

Reviewer #2: (No Response)

**Do you want your identity to be public for this peer review?** For information about this choice, including consent withdrawal, please see our Privacy Policy

Reviewer #1: No

Reviewer #2: No

---

## [Editor Report · Acceptance letter]

PONE-D-25-26588R1

PLOS ONE

Dear Dr. Freihat,

I'm pleased to inform you that your manuscript has been deemed suitable for publication in PLOS ONE. Congratulations! Your manuscript is now being handed over to our production team.

Kind regards,

on behalf of

Dr. Anselm J. M. Hennis

Academic Editor

PLOS ONE